# Intelligent Identification of MoS_2_ Nanostructures with Hyperspectral Imaging by 3D-CNN

**DOI:** 10.3390/nano10061161

**Published:** 2020-06-13

**Authors:** Kai-Chun Li, Ming-Yen Lu, Hong Thai Nguyen, Shih-Wei Feng, Sofya B. Artemkina, Vladimir E. Fedorov, Hsiang-Chen Wang

**Affiliations:** 1Department of Mechanical Engineering and Advanced Institute of Manufacturing with High Tech Innovations, National Chung Cheng University, 168, University Rd., Min Hsiung, Chia Yi 62102, Taiwan; zuoo549674@gmail.com (K.-C.L.); nguyenhongthai194@gmail.com (H.T.N.); 2Department of Materials Science and Engineering, National Tsing Hua University, 101, Sec. 2, Kuang-Fu Road, Hsinchu 30013, Taiwan; mingyenlu@gmail.com; 3Department of Applied Physics, National University of Kaohsiung, 700 Kaohsiung University Rd., Nanzih District, Kaohsiung 81148, Taiwan; swfeng@nuk.edu.tw; 4Nikolaev Institute of Inorganic Chemistry, Siberian Branch of Russian Academy of Sciences, 630090 Novosibirsk, Russia; artem@niic.nsc.ru (S.B.A.); fed@niic.nsc.ru (V.E.F.); 5Department of Natural Sciences, Novosibirsk State University, 1, Pirogova str., 630090 Novosibirsk, Russia

**Keywords:** hyperspectral imagery, deep learning, 3D-CNN, MoS_2_, automated optical inspection

## Abstract

Increasing attention has been paid to two-dimensional (2D) materials because of their superior performance and wafer-level synthesis methods. However, the large-area characterization, precision, intelligent automation, and high-efficiency detection of nanostructures for 2D materials have not yet reached an industrial level. Therefore, we use big data analysis and deep learning methods to develop a set of visible-light hyperspectral imaging technologies successfully for the automatic identification of few-layers MoS_2_. For the classification algorithm, we propose deep neural network, one-dimensional (1D) convolutional neural network, and three-dimensional (3D) convolutional neural network (3D-CNN) models to explore the correlation between the accuracy of model recognition and the optical characteristics of few-layers MoS_2_. The experimental results show that the 3D-CNN has better generalization capability than other classification models, and this model is applicable to the feature input of the spatial and spectral domains. Such a difference consists in previous versions of the present study without specific substrate, and images of different dynamic ranges on a section of the sample may be administered via the automatic shutter aperture. Therefore, adjusting the imaging quality under the same color contrast conditions is unnecessary, and the process of the conventional image is not used to achieve the maximum field of view recognition range of ~1.92 mm^2^. The image resolution can reach ~100 nm and the detection time is 3 min per one image.

## 1. Introduction

The most frequently observed material among all of the two-dimensional (2D) transition metal chalcogenides for the next generation of electronic and optoelectronic components is molybdenum disulphide [1,2,3,4,5,6]. Materials related to nanometer-scale electronic and optoelectronic components, such as field effect transistor [7,8,9,10], prospective memory component [11], light-emitting diode [12,13], and sensors [14,15,16,17,18,19], have been produced due to the excellent spin-valley coupling and flexural and optoelectronic properties of MoS_2_. However, the development of high-performance and large-area characterization techniques has been a major obstacle to the basic and commercial applications of 2D nanostructures.

In the prior art optical film measurement, the atomic force microscope (AFM) has various disadvantages, such as relatively limited scan range and time consuming; thus, it is unsuitable for large-area quick measurements [20,21]. Raman spectroscopy (Raman) is usually only capable of local characterization within the spot, which results in a limited measurement rate; hence, it is unsuitable for large-area analysis. Transmission electron microscopy (TEM) and scanning tunneling microscopy can be characterized at a high spatial resolution of up to atomic scale [22,23]. However, both techniques have the disadvantages of low throughput and complex sample preparation. The use of machine learning, as compared with the abovementioned techniques, in image or visual recognition is a mature application field. The integration of machine learning (SVM, KNN, BGMM-DP, and K-means) with optical microscopes has only begun in recent years. Thus, artificial intelligence has great potential in the recognition of microscopic images, especially nanostructures [24,25]. In 2019, Hong et al. demonstrated the machine learning algorithm to identify local atomic structures by reactive molecular dynamics [26]. In 2020, Masubuchi et al. showed the real-time detection of 2D materials by deep-learning-based image segmentation algorithm [27]. In the same year, Yang et al. presented the identification of 2D material flakes of different layers from the optical microscope images via machine learning-based model [28]. However, the following three shortcomings remain: (1) optical microscope image quality often depends on the user’s experience and will pass through image processing. (2) Only the color space with few feature dimensions will have the possibility of underfitting. (3) Angular illumination asymmetry (ANILAS) of the field of view (FOV) is an important factor that is largely ignored, which results in a certain loss of pixel accuracy [29,30,31].

Here, we used big data analysis and deep learning methods, combined with the hyperspectral imagery common in the remote sensing field, to solve the difficulties that are encountered in the previous literature (e.g., uneven light intensity distribution, image dynamic range correction, and image noise filtering process). The first attempt was made by analyzing the eigenvalues in other dimensions that were not previously used (e.g., morphological features) to improve the prediction accuracy [29,30,31]. An intelligent detection can be achieved to identify the layer number of 2D materials.

## 2. Materials and Methods

### 2.1. Growth Mechanism and Surface Morphology of MoS_2_

The majority of 2D material layer identification studies focus on film synthesis using mechanical stripping [32,33,34]. Although an improved quality of molybdenum disulfide can be obtained, this method cannot synthesize a large-area molybdenum disulfide film with a few layers. The chemical vapor deposition method [35,36,37,38] can produce high-quality and large-area molybdenum disulfide on a suitable substrate surface under a stable gas flux and temperature environment, and it is suitable for current device fabrication [38,39,40]. In 2014, Shanshan et al. explored the sensitivity of the MoS_2_ region growth in a relatively uniform temperature range [41]. In 2017, Lei et al. found that temperature is one of the main factors controlling MoS_2_ morphology [42]. Wei et al. maintained the state for a long time to observe the change of MoS_2_ type when the precursor was heated to a constant temperature interval [43]. In 2018, Dong et al. discussed the nucleation and growth mechanism of MoS_2_ [44]. On the basis of their results, two types of film growth dynamic paths were established: one is a central nanoparticle with a multi-layered MoS_2_ structure and the other is a single-layered triangular dominated or double-layered structure. The conclusion of reference [45] explained the effect of adjusting the growth temperature and carrier gas flux. Understanding the growth pattern mechanism can help us to obtain the initial requirements and judgments of database collection and data tagging.

### 2.2. CVD Sample Preparation

The sample was grown on a sapphire substrate by using CVD to form a MoS_2_ film. The precursor used had Sulfur (99.98%, Echo Chemical Co., Miaoli county, Taiwan) and MoO_3_ (99.95%, Echo Chemical Co., Miaoli county, Taiwan), each placed in the appropriate position of the inner part of the quartz tube. The substrate was placed over the MoO_3_ crucible and in the center of the furnace tube, in a windward position. During growth, different parameters, such as ventilation, heating rate, temperature holding time, and maximum temperature, were set. The MoS_2_ sample was obtained at the end of the growth process. Appendix A shows the schematic of the experimental structure and position of the precursor. Some new pending data indicate the growth of the periodic structure of MoS_2_, which is crucial for the large-scale controllable molybdenum disulfide synthesis and it will greatly benefit the future production of electronic components. In comparison with that of other studies [45,46,47,48,49], this sample was fabricated via laser processing to make periodic holes (hole diameter and depth of approximately 10 μm and 300 nm, respectively), followed by CVD to grow MoS_2_.

### 2.3. Optical Microscope Image Acquisition

MoS_2_ on the sapphire substrate was observed through an optical microscope (MM40, Nikon, Lin Trading Co., Taipei, Taiwan) at 10×, 40×, and 100× magnification rates. For the experimental sample, we recorded the shooting area code (Appendix A to store the images captured via optical microscopy (OM) systematically in the image database. The image had a size of 1600 pixels × 1200 pixels with a depth of 32 bits. The portable network graphics (PNG) file, including the gain image at different dynamic range intervals, was acquired, but color calibration and denoising action were not performed. We aimed to replace the cumbersome image processing with deep learning. On the basis of the amount of data that were collected in this experiment, approximately 90 pieces of 2 × 2 cm^2^ growth samples were obtained, and ~2000 images were used as sources of data exploration.

### 2.4. System Equipment and Procedures

This study aims to construct a system for the automated analysis of different layers of MoS_2_ film. The system architecture is mainly divided into four parts, as shown in Appendix A. The program flow is as follows: (1) database: the prepared molybdenum disulfide sample is measured using a Raman microscope to determine the position of each layer distribution, and an image is taken using an optical microscope and CCD to capture the same position (Image Capture System). (2) Offline Training: the obtained CCD image is combined with hyperspectral imaging technology (VIS-HSI) to convert the spectral characteristics of each layer of molybdenum disulfide, and data preprocessing is performed. (3) Model Design: the data are further trained in deep learning, thereby completing the establishment of our classification algorithm. (4) Online Service: when a new sample of molybdenum disulfide is to be analyzed, it is placed under an optical microscope to capture the surface image by using CCD. Subsequently, the spectral characteristic value of each pixel is obtained through the hyperspectral imaging technique, and the model is predicted by training. The number of layers of the molybdenum disulfide film at the pixel position is determined, and the molybdenum disulfide film with different layers in the image is visualized while using different colors.

### 2.5. Tag Analysis and Feature Extraction Workflow

Figure 1 shows the processing step before the data enters the model. Layer number labeling of the manual area circle mask (Mask) is performed and the data are divided when the captured image is converted into a spectrum image by using a hyperspectral image technology; we will measure the result on the basis of the Raman spectrum (Appendix A) [50,51,52,53,54]. The categories are substrate, monolayer, bilayer, trilayer, bulk, and residues, which are our ground truths. Two types of data are available for model training, namely “feature” and “label”. Feature has two types: one is hyperspectral vector as input for deep neural network (DNN) and one-dimensional (1D) convolutional neural network (1D-CNN), and the other is spatial domain hyperspectral cube as input for three-dimensional (3D) convolutional neural network (3D-CNN). After data preprocessing, we divide the dataset into three parts. We randomly select 80% of the labeled samples as training data, 20% as the verification set, and the remaining unmarked parts for the test set. As the light intensity distribution is incompletely covered in the dataset in the training and validation sets, we need a test set to help us measure whether the model has this capability.

### 2.6. Visible Hyperspectral Imaging Algorithm

The VIS-HSI used in this study is a combination of CCD (Sentech, STC-620PWT) and visible hyperspectral algorithm (VIS-HSA). The calculated wavelength range is 380–780 nm and the spectral resolution is 1 nm. The core concept of this technology is to input the image captured by the CCD in the OM to the spectrometer, such that each pixel of the captured image has spectrum information [55]. Appendix A shows the flow of the technology, while using MATLAB. A custom algorithm is created.

### 2.7. Data Preprocessing and Partitioning

All of the MoS_2_ samples were obtained at different substrate locations and uneven illumination distribution to form a dataset. Two types of features were available: one is the 1 × 401 hyperspectral vector central pixel, which extracts the marker data as DNN and 1D-CNN inputs, and the other is the 1 × 5 × 5 × 401 hyperspectral cube, which regards the spatial region as the center of the cubes. The category labels of the pixels extracted the tag data as the 3D-CNN input. With single hyperspectral image data as an example (Appendix A), the marked area was divided into 80% training set and 20% validation set, whereas the unmarked area was the test set. Finally, the new pending data were generated. The generalization capability of the model was evaluated. The training set part had oversampling and data enhancement for the hyperspectral cube, because it contained spatial features and it was processed via horizontal mirroring and left and right flipping.

### 2.8. Software and Hardware

The model usage environment was implemented in the Microsoft Windows 10 operating system while using TensorFlow Framework version 1.10.0 and Python version 3.6. Open-source data analysis platforms, namely, Jupyter notebook, SciPy, NumPy, Matplotlib, Pandas, Scikit-learn, Keras, and Spectral, were used to analyze the feature values. The training hardware used a consumer-grade desktop computer with a GeForce GTX1070 graphics card (NVDIA, Hong Kong, China) and Core i5-3470 CPU @ 3.2GHz (Intel, Taipei, Taiwan).

## 3. Results

The model must have a certain recognition capability, because ensuring that the quality of each captured image is the same is impossible. Therefore, we add several image quality features to the hyperspectral image data of different imaging qualities to make the model close to the practical classification performance. We will explore the models with different magnification rates in order to discuss the spatial resolution and mixed pixel issues [56,57].

### 3.1. Model Framework for Deep Learning

The classification prediction model uses three models, namely, DNN, 1D-CNN, and 3D-CNN, as shown in Figure 2. Among the model parameters, the learning rate is adjusted to 1 × 10^−6^ to 5 × 10^−6^ and the batch size (batch) is based on the difference between the model and data. The size and dropout are 24 and 0.25, respectively, and the selected optimizer is RMSprop. Figure 2a illustrates a schematic of the basic DNN model architecture. The input is a hyperspectral vector feature belonging to single-pixel spectral information. The model only contains three layers of fully connected layer. Additional outer neuron nodes are expected to extract relatively shallow features [58]. The six categories that we classify are the outputs. Figure 2b displays a schematic of the 1D-CNN model architecture. The input is consistent with the DNN model. The model consists of four convolutional layers, which include two pooling layers and two fully connected layers. CNN convolution kernel has rights-sharing characteristics. We believe that convolving of spectral features is equivalent to chopping different frequencies. Given the 1 nm resolution of the band, the correlation with adjacent feature points is high. The pooling layer can help to eliminate features that are too similar in the neighborhood and reduce the redundant dimensions of features [59]. Figure 2c shows a schematic of the 3D-CNN model architecture. The input belongs to a hyperspectral cube type of the space-spectral domain. It extracts a feature cube consisting of pixels as d × d × N in a small spatial neighborhood (not the entire image) along the entire spectral band as input data and convolves with the 3D kernel to learn the spectral spatial features. The reason for using neighboring pixels is based on the observation that pixels in the small spatial neighborhood often reflect similar characteristics [60]. This observation is proven in Ref [55,61], which indicated that the small 3 × 3 core is the best option for spatial features; thus, only two convolution operations are performed and the sample space size is set to 5 × 5, which can be reduced to only two convolutional layers (1 × 1). The spatial domain of each layer is extracted. The first 3D convolutional layers, namely, C1 and C2, each contain a 3D kernel. The kernel size is $K_{1}^{1}$ × $K_{2}^{1}$ × $K_{3}^{1}$, resulting in two sizes of 3D feature cubes as (d − $K_{1}^{1}$ + 1) × (d − $K_{2}^{1}$ + 1) × (N − $K_{3}^{1}$ + 1). Two 3D feature cubes, namely, C1 and C2, as (d − $K_{1}^{1}$ + 1) × (d − $K_{2}^{1}$ + 1) × (N − $K_{3}^{1}$ + 1), are used as inputs. The second 3D convolutional layer, namely, C3, involves four 3D cores (with a size of $K_{1}^{2}$ × $K_{2}^{2}$ × $K_{3}^{2}$) and produces eight 3D data cubes, each as (d − $K_{1}^{1}$ − $K_{1}^{2}$ + 2) × (d − $K_{2}^{1}$ − $K_{2}^{2}$ + 2) × (N − $K_{3}^{1}$ − $K_{3}^{2}$ + 1) [62].

### 3.2. Training Results under Three Deep Learning Models with 10× Magnification

Figure 3 shows the calculation results of the sapphire substrate sample via three models, namely, DNN, 1D-CNN, and 3D-CNN, under 10× magnification. Figure 3a–c exhibit the convergence curve and training time of the loss and accuracy in the three algorithms, respectively. 3D-CNN has a longer training time and epoch than the first two models, and its input feature variable has more space domain parts; thus, it takes more time to start the convergence process. Figure 3d–f display the results of the confusion matrix of the verification set in the three algorithms, respectively. Table 1 presents the evaluation results of each category. Classifier precision determines how many of the positive categories of all the samples are true positive. The recall rate indicates how many of the true-positive category samples are judged as positive category samples by the classifier. F1-score is the harmonic mean of the accuracy and recall rate. Macro-average refers to the arithmetic mean of each statistical indicator value of all categories. The micro-average is used to establish a global confusion matrix for each model example in the dataset without category and then calculate the corresponding indicators. 3D-CNN accuracy is better from the indicators. Appendix A exhibit the remaining training procedures at 40× and 100× magnification rates. Table 1 and Appendix A show that the evaluation results under a small magnification are relatively poor. Therefore, the mixed pixels may cause the pixels to contain additional mixing or noise factors.

### 3.3. Prediction Results at 10× Magnification

Figure 4 presents a randomly selected sample at 10× magnification, which is predicted by three models. Figure 4a displays the OM image under the corresponding range of the prediction data. Figure 4e exhibits the OM image of new pending data. Figure 4b–d show the training data and Figure 4f–h present the prediction results for the new data (new pending data) under three models (color classification image), respectively. The results from the region of interest (ROI) in the training data indicate that the DNN and 1D-CNN models cannot accurately predict the damaged region when the MoS_2_ film encounters external force damage and the crystal structure is missing. 3D-CNN can be clearly judged under the destruction of the region, and Appendix A exhibits the remaining new pending data predictions. In other predictions for 40× and 100× magnification rates (Appendix A), the color classification image (false-color composite) can be easily found for each model, but it will be reduced in the opposite FOV detection range.

### 3.4. Differences in Models at Three Magnification Rates

In this experiment, the classification algorithm is based on the pixel unit data in the image; thus, determining how to obtain the quantitative value of the enhanced precision for the OM is crucial. The majority of existing microscopes achieve uniform spatial irradiance through Köhler illumination [63]. However, some shortcomings remain in the need for quantitative measurement and analysis, for example, FOV nonuniform illumination asymmetry (ANILAS) [64,65] is a largely important factor that is ignored (Appendix A). This phenomenon leads to a certain loss in pixel accuracy. Thus, we attempt to understand the various feature data types through deep learning in the case of uneven illumination distribution.

The FOV sizes of the 10×, 40×, and 100× magnification rates are 1.6 mm × 1.2 mm, 0.4 mm × 0.3 mm, and 0.17 mm × 0.27 mm, respectively, which are the actual detection sizes at the time of prediction. Figure 5 presents the optimal loss values of the (a) training and (b) verification sets for three models at three magnification rates. The optimal train loss is higher than the validation loss, partly because of the use of data enhancements in the training set, which makes the model difficult to learn when the data are increased in diversity. Figure 5 shows that the three models have low loss values under 100× magnification. When considering the difference in spatial resolution, the pixel resolutions at 10×, 40×, and 100× magnification rates are 0.5, 0.25, and 0.1 μm, respectively. Therefore, the problem of mixed pixels is further serious at a small magnification. 3D-CNN is the best among the three models at different magnification rates.

As 3D-CNN demonstrates the best generalization capability at different magnification rates, we will only discuss the results of this algorithm. Figure 6a shows an OM image of a large-area periodically grown single-layer MoS_2_ on a sapphire substrate (also defined as ROI-3). The actual corresponding size is 1 × 1 mm, and the microscope magnification is 10×. We can observe that a single layer of MoS_2_ is distributed in a star shape around the hole. Figure 6a–f display the analysis of the color classification image (false-color composite). Figure 6b–d present the OM images of the predicted results under three magnification rates, namely, 100× (ROI-1), 40× (ROI-2), and 10× (ROI-3), respectively.

We obtain the magnified images of the ROI-4 and ROI-5 regions from the color classification image under 10× magnification (Figure 6d) as Figure 6e,f, respectively. Corresponding to other magnification rates (Figure 6b,c) in the same region of the color classification image, we observe that a small magnification will be limited by the spatial resolution. Consequently, the fine type of features will be blurred or impurity points are further difficult to identify. Appendix A discusses the probability of class prediction confidence in images with various magnification rates. This finding is different from previous research arguments [66]. Previous studies have considered that the poor image quality is due to several noise points and the surrounding blur at a large magnification or that fine impurities are caused by deposition, resulting in re-traditional classification algorithms. The effect is not good, but the small impurities will be ignored when the spatial resolution is low. However, in the present study, the cognition of the model in deep learning solves the bottleneck of the previous problem.

### 3.5. Instrument Measurement Verification

In the new pending data section, we observe the accuracy of the samples from Raman spectroscopy mapping, as shown in Appendix A. We show two oscillation modes of in-plane ($E_{2g}^{1}$) and out-of-plane ($A_{1g}$) in Raman mapping analysis. The classification of the number of layers is determined by two peak differences. Appendix A presents the SEM measurement results. The instrument can be judged by the gray level. However, the instrument measurement cannot determine the number of layers and it can only be observed from relative contrast. The PL spectroscopy results (Appendix A) indicate that the mapping diagram is either 625 or 667 nm, and the periodic growth of the single-layer to multilayer distribution of MoS_2_ has good uniformity. In Appendix A, the material of the sample profile is divided via HRTEM [67,68].

## 4. Conclusions

This study is aimed at the layer number discrimination of molybdenum disulfide film on sapphire. In comparison with the current measurement instruments, such as Raman, SEM, AFM, and TEM, the proposed equipment has a large detection area, less time, and low cost. The equipment can detect the low number of layers of molybdenum disulfide film. Unlike in the past, we use deep learning, but not image processing, for analysis, and experimentally confirm that 3D-CNN has the best precision and generalization capability. The reason is that the 3D-CNN model initially adds the spatial domain of the morphological features of MoS_2_ to learn to avoid the misjudgment caused by the difference in imaging quality due to noise or to make further accurate judgments on the fuzzy regions between the category regions. For the problem that the low magnification is limited by the spatial resolution, which results in fine contaminants and edge blur morphology, the GAN model can be used to achieve the super-resolution method in low magnification [69,70]. In future research, we hope to integrate all types of 2D materials and various substrates, such as in the case of heterogeneous stack, in order to easily distinguish different 2D materials and their different layers easily. The future ideas are to determine the inference of the growth pattern of MoS_2_ by detecting the image in real time and to avoid machine termination for reducing time and related costs under impurity intrusion

## Figures and Tables

**Figure 1 nanomaterials-10-01161-f001:**
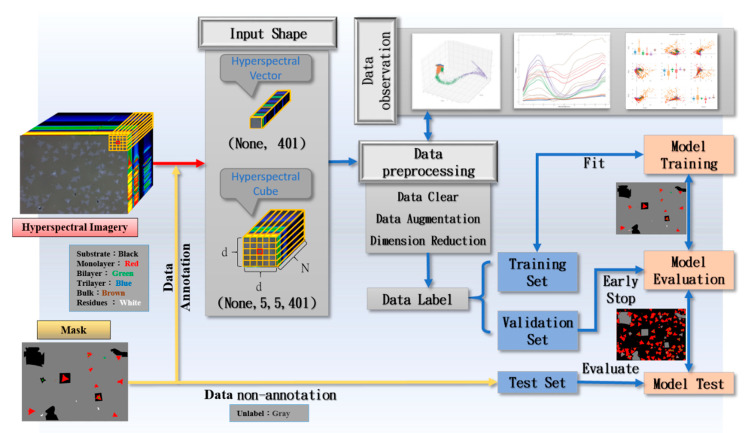
Data feature and label processing.

**Figure 2 nanomaterials-10-01161-f002:**
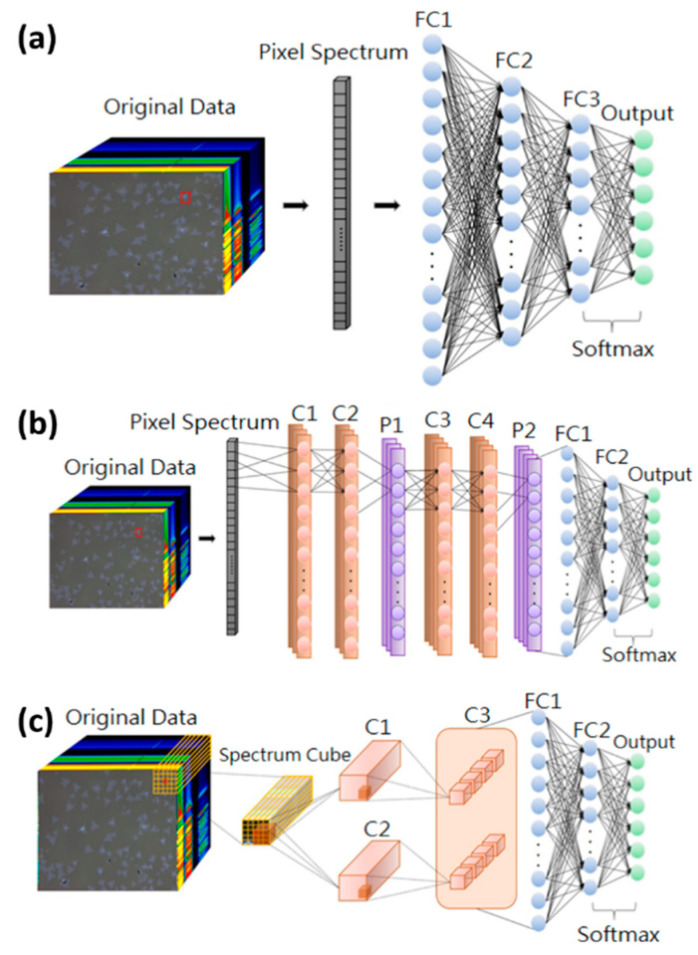
Schematic of the model construction. (**a**) Deep neural network (DNN), (**b**) one-dimensional (1D) convolutional neural network (1D-CNN), and (**c**) three-dimensional (3D) convolutional neural network (3D-CNN), where the inputs in (**a**,**b**) are hyperspectral vectors, and the input in (**c**) is hyperspectral cube. For the outputs of the three models, Softmax is used as a classifier for six categories: substrate, monolayer, bilayer, trilayer, bulk, and residues.

**Figure 3 nanomaterials-10-01161-f003:**
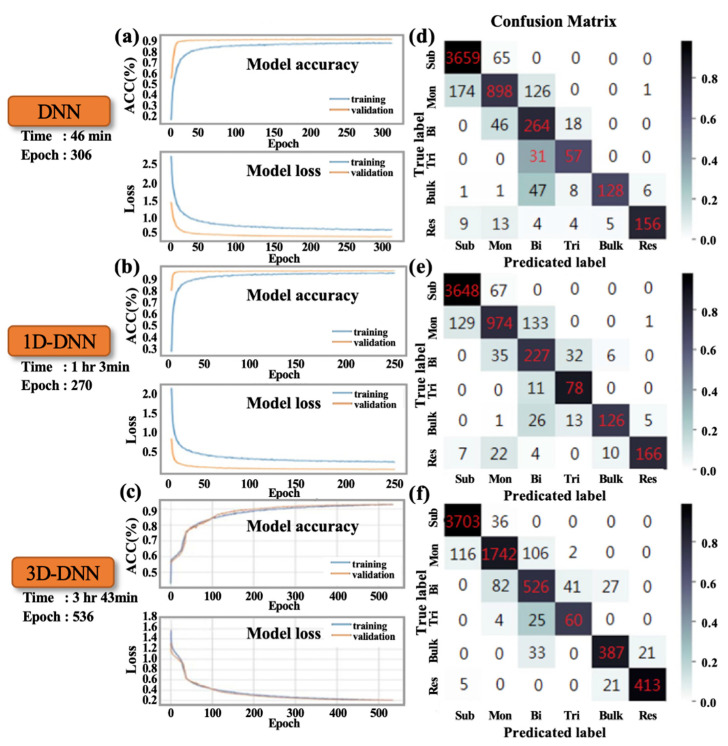
At 10× magnification condition: accuracy (ACC) and loss in the convergence process in (**a**) DNN, (**b**) 1D-CNN, and (**c**) 3D-CNN; confusion matrix results of the validation set in (**d**) DNN, (**e**) 1D-CNN, and (**f**) 3D-CNN.

**Figure 4 nanomaterials-10-01161-f004:**
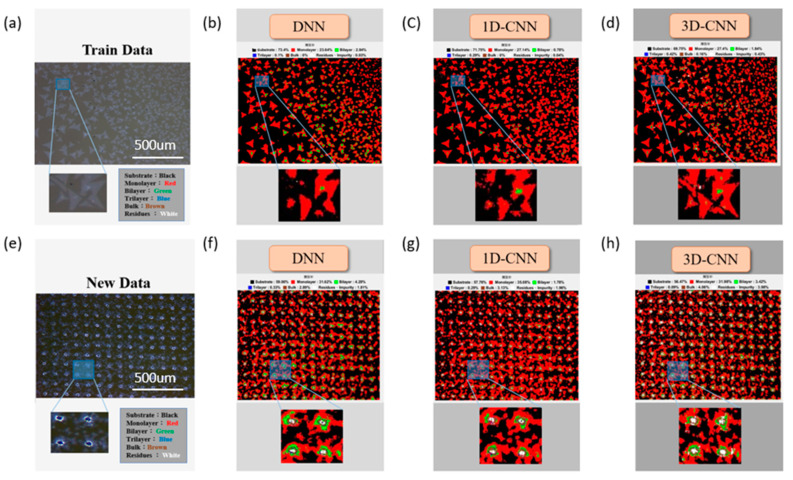
At 10× magnification: (**a**,**e**) optical microscopy (OM) images of the training (train data) and new test (new pending data) data, respectively; predicted results of the color classification image (false-color composite) under three models for the training data (**b**–**d**), namely, (**b**) DNN, (**c**) 1D-CNN, and (**d**) 3D-CNN, and for the new pending data (**f**–**h**), namely, (**f**) DNN, (**g**) 1D-CNN, and (**h**) 3D-CNN.

**Figure 5 nanomaterials-10-01161-f005:**
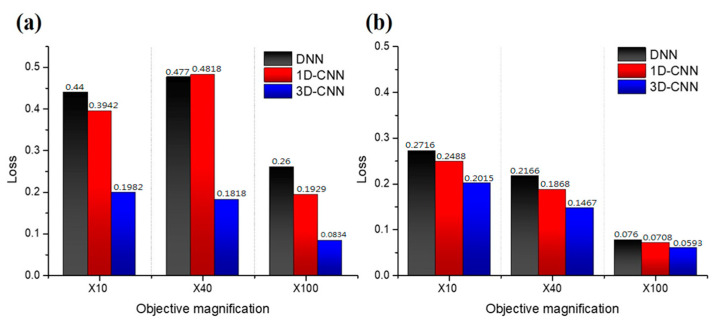
Optimal loss of the three models in the (**a**) training and (**b**) validation sets at 10×, 40×, and 100× magnification rates.

**Figure 6 nanomaterials-10-01161-f006:**
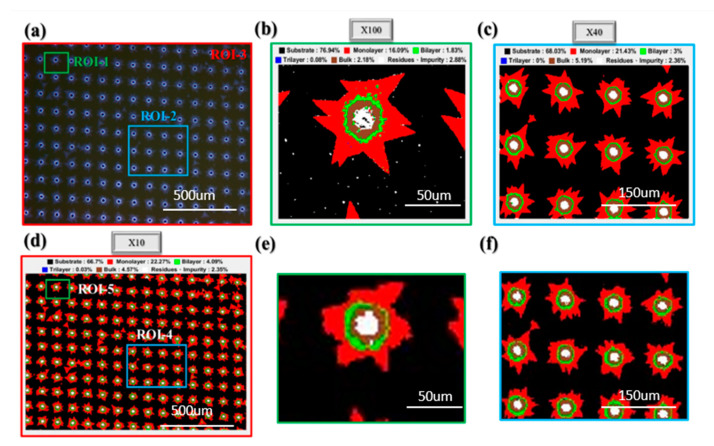
(**a**) OM images of the large-area periodic growth of single-layer MoS_2_ on sapphire substrates at (**b**) 100× (ROI-1), (**c**) 40× (ROI-2), and (**d**) 10× (ROI-3) magnification rates. Color classification map predicted at 10× magnification: (**e**,**f**) are the amplification results of ROI-4 and ROI-5 ranges, respectively, corresponding to ROI-1 and ROI-2 circle selection.

**Table 1 nanomaterials-10-01161-t001:** Model evaluation indicators for the three models at 10× magnification.

Final Accuracy (Validation Data)									
	DNN			1D-CNN			3D-CNN		
	Precision	Recall	F1-Score	Precision	Recall	F1-Score	Precision	Recall	F1-Score
**Substrate**	0.9521	0.9825	0.9671	0.9641	0.9820	0.9729	0.9684	0.9904	0.9792
**Monolayer**	0.8778	0.7490	0.8083	0.8863	0.7874	0.8339	0.9345	0.8861	0.9097
**Bilayer**	0.5593	0.8049	0.6600	0.5661	0.7567	0.6476	0.7623	0.7781	0.7701
**Tri-layer**	0.6552	0.6477	0.6514	0.6341	0.8764	0.7358	0.6504	0.7339	0.6897
**Bulk**	0.9624	0.6702	0.7901	0.8873	0.7368	0.8051	0.8897	0.8776	0.8836
**Residues**	0.9571	0.8168	0.8814	0.9651	0.7943	0.8714	0.9516	0.9408	0.9462
**micro average**	0.9023	0.9023	0.9023	0.9123	0.9123	0.9123	0.9296	0.9296	0.9296
**macro average**	0.8273	0.7785	0.7930	0.8172	0.8223	0.8111	0.8595	0.8678	0.8631
**weighted average**	0.9100	0.9023	0.9026	0.9190	0.9123	0.9134	0.9300	0.9296	0.9295

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
