# Peer review of "Intelligent Identification of MoS_2_ Nanostructures with Hyperspectral Imaging by 3D-CNN"

_nanomaterials, 2020, doi:10.3390/nano10061161_

Round 1
Reviewer 1 Report
1.
The task of the model is to select one of six classes. It is known that not all classes may have a similar number of cases in a training set and in a validation set. However, I wonder if you can expect the correct recognition of the “trilayer” class, for example. This class is represented very rarely. This problem is also visible in the confusion matrix. In my opinion, these problems require the authors' comment.
2.
The softmax function was used in the output layer of the neural network. The softmax function is an exponential function with a value additionally normalized in such a way that the activation sum for the whole layer equals 1. If the softmax-type function is used in the output layer of a neural network designed for solving classifications tasks, it allows one to interpret the activation level of output layer neurons as an estimated probability of membership in a particular class. Are the differences in the activation values of neurons large in incorrectly classified cases? Of course, the problem is difficult to analyze due to the large number of classifications. Fig. S14 provides interesting information on this topic. Maybe it would be interesting to briefly discuss this problem?
3.
Table 1 shows the model evaluation indicators. The authors define the classifier precision and the recall. In my opinion, the difference between these indicators is not clearly explained. It is also not explained how and for what purpose the weighted average is calculated.
4.
lines 255 and 256
"The FOV sizes of the 10×, 40×, and 100× magnification rates are 0.17 mm×0.27 mm, 0.4 mm×0.3 and 1.6 mm×1.2 mm, respectively, which are the actual detection sizes at the time of prediction."
Is this information correct?
5.
In Fig. 4 there are various descriptions of the y axis in the model accuracy chart. I mean ACC (%) or accuracy. Is the percentage unit correct in this case? The comment also applies to Figs. S11 and S12.
6.
Some abbreviations are explained too late. For example, the abbreviation OM was used for the first time on line 100 and it was explained on line 248.
7
Some of the information shown in the figures in the supplement are difficult to read. This comment applies especially to the Figure S11 and Figure S12.
Reviewer 2 Report
The authors present an analysis on using different deep learning models to automatically discriminate the number layers in the case of a 2D material, namely molybdenum disulfide film, on sapphire. In my view addressing new ways of combining artificial intelligence with materials science is a huge scientific priority, given that these methods can significantly augment the outputs of traditional materials' characterization approaches. However, I find this manuscript to be very modest in conveying to the reader useful ideas. Besides the poor use of English language, the writing style is chaotic. An example that very well reflects this is the abstract, where the authors omit nominating the material that they focus one, or the characterization techniques that they use. Furthermore, in the manuscript the authors do not present how they compare to the state of the art, if similar applications have been reported, and they do not accurately describe what their method brings new. I have carefully read the manuscript several times, but each time I find it even more confusing. I would suggest to the authors to completely restructure their manuscript and present in a logical order: what problem they want to solve, what is the state of the art on solving that problem, what their method brings new, and then present the technical results in a logically ordered manner.
Reviewer 3 Report
Review of “Intelligent Identification of MoS2 Nanostructures with Hyperspectral Imaging by 3D-CNN” written by Kai-Chun Li, Ming-Yen Lu, Hong Thai Nguyen, Shih-Wei Feng, Sofya B. Artemkina, Vladimir E. Fedorov and Hsiang-Chen Wang, manuscript submitted to Nanomaterials.
The authors report a series of measurements on MoS2 nanostructures grown on sapphire surfaces. The paper discuss the creation, training, and exploitation of artificial neural networks to classify pixels in optical images by the number of atomic layers in the MoS2 nanostructures. The training of the NN is based on the ability to classify the structures by recording on the same sample areas Raman spectra maps that are sensitive to the number of atomic layers. Overall this manuscript is well written and the discussion sensible. The methodology of data processing, at the core of the paper, is well described which is very useful. I have a few comments below that the authors should consider during revision.
- The abstract states that: “The resolution can reach ca. 100 nm,…”. The statement suggests that the authors are considering here the spatial resolution. This statement is not well addressed in the manuscript, if at all. The microscopy images are also not scaled throughout the manuscript so that it is unclear what is actually the spatial resolution. The authors also did not mention the NA of the objectives used (nor the immersion media if any). 100 nm resolution would suggest some form of super-resolution and would need to be discussed further. The scale of the optical images should be indicated. One could also question what is the scale of the MoS2 Are these nano or micro?
- The same statement continues “… the detection time is 30 images/sec”. The statement would probably need to be better described in the manuscript. Is this the CCD shutter time? Does that time match or include the data processing as well? I suppose the authors are stating here that the method is overall fast versus Raman mapping (this is clearly the case of course) but it would be informative to know what is this time exactly. Following that, would the authors expect that the NNs need to be retrained say when changing sample or when the illumination changes, etc and if so how would the “complete” detection (with processing) scales with Raman for example.
- From their optical images, recorded with a CCD, the authors build hyperspectral images. This is fine (maybe the methodology for this could be more detailed) but at the same time it is unclear why this is needed and the authors should better indicate why they prefer to add this step in the pre-processing. From the manuscript, I understand that the CCD reading (3 colors) is converted into a spectrum, and the spectrum (including those from adjacent pixels when the larger 3D-CNN is used) is then the input for the NNs. This operation (i.e., 3 colors into hyperspectrum) does not seem to add any information and one would (perhaps naively) think that the 3 color values could be used directly as entry of the NNs. Could the authors expend on this in a revision?
- The authors are compiling a large amount of figures and data. This also leads to many labels in the figures that are not readable as too small. The authors might want to address that.
Round 2
Reviewer 2 Report
The modifications implemented by the authors following my previous comments are welcome, and consolidate the structure of the manuscript.
It would have been useful if their work is better placed into a context. Unlike the authors state in their rebuttal letter, other works combining deep learning with MoS2 characterization exist. I would encourage them to briefly discuss these previous efforts, and highlight potential overlap or complementarities with their work. Some of these previous works that I find relevant are listed below, but others should be considered as well.
10.1038/s41699-020-0137-z
10.1021/acs.jpclett.9b00425
10.1016/j.jmat.2019.03.003
/10.1016/j.eml.2020.100771
